# Glycan Epitopes on 201B7 Human-Induced Pluripotent Stem Cells Using R-10G and R-17F Marker Antibodies

**DOI:** 10.3390/biom11040508

**Published:** 2021-03-29

**Authors:** Yuko Nagai, Hiromi Nakao, Aya Kojima, Yuka Komatsubara, Yuki Ohta, Nana Kawasaki, Nobuko Kawasaki, Hidenao Toyoda, Toshisuke Kawasaki

**Affiliations:** 1Laboratory of Bio-analytical Chemistry, College of Pharmaceutical Sciences, Ritsumeikan University, Shiga 525-8577, Japan; ph0011vi@gmail.com (Y.N.); sano.a1029@gmail.com (A.K.); ph0104ev@ed.ritsumei.ac.jp (Y.K.); hidenao@ph.ritsumei.ac.jp (H.T.); 2Glycobiotechnology Laboratory, Ritsumeikan University, Shiga 525-8577, Japan; 15v00381@gst.ritsumei.ac.jp (H.N.); 14v00048@gst.ritsumei.ac.jp (N.K.); 3Department of Medical Life Science, Yokohama City University, Kanagawa 230-0045, Japan; yuki.ohta@hotmail.com (Y.O.); nana@yokohama-cu.ac.jp (N.K.)

**Keywords:** human-induced pluripotent stem cells (hiPSCs), monoclonal antibodies, R-10G, R-17F, keratan sulfate, podocalyxin, keratanase II, endo-β-galactosidase

## Abstract

We developed two human-induced pluripotent stem cell (hiPSC)/human embryonic stem cell (hESC)-specific glycan-recognizing mouse antibodies, R-10G and R-17F, using the Tic (JCRB1331) hiPSC line as an antigen. R-10G recognizes a low-sulfate keratan sulfate, and R-17F recognizes lacto-N-fucopentaose-1. To evaluate the general characteristics of stem cell glycans, we investigated the hiPSC line 201B7 (HPS0063), a prototype iPSC line. Using an R-10G affinity column, an R-10G-binding protein was isolated from 201B7 cells. The protein yielded a single but very broad band from 480 to 1236 kDa by blue native gel electrophoresis. After trypsin digestion, the protein was identified as podocalyxin by liquid chromatography/mass spectrometry. According to Western blotting, the protein reacted with R-10G and R-17F. The R-10G-positive band was resistant to digestion with glycan-degrading enzymes, including peptide N-glycanase, but the intensity of the band was decreased significantly by digestion with keratanase, keratanase II, and endo-β-galactosidase, suggesting the R-10G epitope to be a keratan sulfate. These results suggest that keratan sulfate-type epitopes are shared by hiPSCs. However, the keratan sulfate from 201B7 cells contained a polylactosamine disaccharide unit (Galβ1-4GlcNAc) at a significant frequency, whereas that from Tic cells consisted mostly of keratan sulfate disaccharide units (Galβ1-4GlcNAc(6S)). In addition, the abundance of the R-10G epitope was significantly lower in 201B7 cells than in Tic cells.

## 1. Introduction

In the last decade, many interesting findings have been reported in the field of stem cells. Human-induced pluripotent stem cells (hiPSCs) are already used as tools for drug development and disease modeling, and their most important potential application is the generation of cells and tissues that can be used for regenerative medicine [1]. Stem cell research is obviously one of the most fascinating and exciting areas of contemporary biology, including glycobiology [2].

Glycans attached to proteins (glycoproteins) and lipids (glycolipids) are mainly located at the outermost surface of the cell. Significant alterations in the cellular glycoform may occur during differentiation and glycans could serve as specific stem cell markers and cellular differentiation biomarkers. At the time when hiPSCs were first generated from differentiated cells, SSEA4 [3,4] and TRA1-60 [5] were widely used as pluripotent cell markers. However, these traditional marker antibodies were generated against human teratocarcinoma cells: 2102Ep for TRA-1-60 and GCTM2 for SSEA-4. In other words, these antibodies are not specific to hiPSCs/hESCs; rather, they recognize those glycans that are common to hiPSCs/hESCs and human embryonal carcinoma cells (hECs). New antibodies that are capable of distinguishing between malignant and normal phenotype would be valuable. With this background, we generated antibodies by using the Tic (JCRB1331) hiPSC line as an antigen [6]. We first selected hiPSC (Tic)-positive hybridomas, from which hEC (2102Ep)-positive hybridomas were excluded. Most of the antibodies raised recognized both hiPSCs (Tic) and hECs (2l02Ep). But, fortunately, we obtained R-10G and R-17F antibodies that were able to bind nearly specifically to hiPSC/hESC. The R-10G binding protein was isolated from Tic cell lysates using an R-10G affinity column and identified as podocalyxin by liquid chromatography–tandem mass spectrometry (LC-MS/MS) after SDS-PAGE [6]. The minimum epitope structure of R-10G was indicated to be Galβ1-4GlcNAc(6S) β1-3 Galβ1-4GlcNAc(6S) β1 with ELISA by using synthetic oligosaccharides [7]. The R-17F antigen was initially isolated to homogeneity from total lipid extracts of Tic iPSC and identified as Fucα1-2Galβ1-3GlcNAβ1-3Galβ1-4Glc-Cer by matrix-assisted laser desorption ionization–time of flight mass spectrometry [8]. Subsequently, R-17F epitope was also found to be expressed on the R-10G-binding protein [9]. On the other hand, SSEA4 recognizes Neu5Ac α 2-3Gal β 1-3GalNAcβ1-3Gal-R [3] and the major epitope of TRA1-60 is Galβ1-3GlcNAcβ1-3Galβ1-4GlcNAcβ1 [7,10]. Thus, the glyco-epitopes of R-10G and R-17F antibodies are totally different from those of SSEA4 and TRA1-60. The immuno-histochemical studies by laser confocal microscopy and flowcytometry demonstrated that R-10G and R-17F epitopes were expressed not only on two hiPSCs lines, Tic and 201B7, but also on two hESCs lines, H9 and KhES-3 [6,8], suggesting that R-10G and R-17F epitopes can be marker molecules on the undifferentiated stage of hiPSCs. Another important observation was that the R-10G and R-17F epitopes are localized differentially on these cells: R-10G epitope appears to be distributed evenly on the whole surface of cells, while R-17F epitope appears to be selectively localized on cell membranes.

The immuno-histochemical profiles described above strongly suggested that there are high similarities in glycan profiles on the surfaces of hiPSCs/hESCs. The present study aims to dissect these similarities from several aspects using 201B7hiPSC as a basis for comparison, which is a prototype of hiPSCs generated by Dr. Yamanaka [1] and is currently used as a kind of control or standard hiPSC. Subjects to be explored include: identification of specific glycoproteins carrying multiple glycan-epitopes, type of glycoprotein carrying major glyco-epitopes (N-glycans, O-glycans, or glycosaminoglycans), and biochemical characterization of keratan sulfate, which is an actively developing area in glycobiology.

## 2. Materials and Methods

### 2.1. Antibodies and Enzymes

Anti-TRA-1-60 (clone TRA-1-60, mouse IgM) was obtained from R&D Systems Inc. (Minneapolis, MN, USA). Horseradish peroxidase (HRP)-conjugated rabbit anti-mouse Ig was obtained from Agilent Technology (Santa Clara, CA, USA). Polyclonal goat anti-podocalyxin IgG and HRP-conjugated rabbit anti-goat IgG secondary antibodies were obtained from R&D Systems. Anti-human iPSC/ESC, R-10G (mouse IgG1), and R-17F (mouse IgG1) antibodies were prepared as described previously [6,8].

Peptide N-glycanase (PNGase F; recombinant protein from *Escherichia coli*) was obtained from Roche Diagnostics GmbH (Mannheim, Germany), neuraminidase (*Arthrobacter ureafaciens*) from Nacalai Tesque (Kyoto, Japan), α1-3/4 fucosidase from TaKaRa Bio, Inc. (Shiga, Japan), and α1-2 fucosidase from New England Biolabs (Ipswich, MA, USA). Chondroitinase ABC (*Proteus vulgaris*), heparinase mixture (heparinase, heparitinase I, and heparitinase II), keratanase (*Pseudomonas* sp.), keratanase II (*Bacillus* sp.), and endo-β-galactosidase (*Escherichia freundii*) were obtained from Seikagaku Biobusiness (Tokyo, Japan).

### 2.2. Cell Culture 

The hiPSC line Tic (JCRB1331) was obtained from the Japanese Collection of Research Bioresources (JCRB) Cell Bank, National Institutes of Biomedical Innovation, Health and Nutrition (Osaka, Japan). 201B7 (HPS0063) cells were obtained from the Center for iPS Cell Research and Application, Kyoto University (Kyoto, Japan). These cells were prepared after transfection of four defined factors (Oct3/4, Sox2, Klf4, and c-Myc) [1,6]. The cells were maintained in Knockout Serum Replacement medium (Invitrogen-Life Technologies, Carlsbad, CA, USA) on mitomycin C-inactivated mouse embryonic fibroblasts (Merck Millipore, Billerca, MA, USA) and harvested by treatment with 0.1% ethylenediaminetetraacetic acid (EDTA)-4Na/phosphate-buffered saline, as described previously [6].

### 2.3. Isolation of R-10G-Binding Protein from 201B7 Cells

hiPSC lysates were prepared by sonicating 201B7 cells (2.4 mg protein/2.4 × 10^7^ cells, pooled cells after 9, 11, and 15 numbers of passages) in complete radioimmunoprecipitation assay (RIPA) buffer (1.25 mL; 6 mM Tris-HCl (pH 8.0), 150 mM NaCl, 1% Nonidet P-40, 0.5% sodium deoxycholate, 0.1% SDS, 0.004% sodium azide) supplemented with phenylmethylsulfonyl fluoride, sodium orthovanadate, and protease inhibitor cocktail (Santa Cruz Biotechnology, Inc., Santa Cruz, CA, USA). The lysates were centrifuged to remove insoluble residues, and the supernatant was added to an R-10G Sepharose 4B column (gel volume, 0.4 mL), which had been prepared by coupling R-10G (4 mg protein) to BrCN-activated Sepharose 4B (1.0 mL; GE Healthcare, Tokyo, Japan) in 0.1 M NaHCO_3_ buffer (pH 8.3)/0.5 M NaCl according to the manufacturer’s instructions. After washing the column with complete RIPA lysis buffer, bound proteins were eluted in elution buffer consisting of 10 mM Tris-HCl (pH 7.4), 150 mM NaCl, 0.1 M diethylamine-HCl (pH 11.5), and 0.1% Nonidet P-40. The eluate containing R-10G antigen was collected in microtubes (200 µL/tube) and immediately neutralized by adding 1 M Tris-HCl buffer (pH 6.8) (40 µL/tube). 

### 2.4. SDS-PAGE and Western Blotting

SDS-PAGE and Western blotting were performed according to the methods of Laemmli [11] and Towbin [12], respectively. Briefly, samples were resolved by 4–15% gradient SDS-PAGE (Mini-PROTEAN TGX gel; Bio-Rad, Hercules, CA, USA) under reducing conditions, followed by Western blotting or protein staining. For Western blotting, resolved proteins were transferred to Immobilon transfer membranes (Millipore, Billerica, MA, USA), followed by detection using specific Abs. For visualization, Immunostar Zeta (Fujifilm Wako Pure Chemical Corp., Osaka, Japan) was used with HRP-conjugated rabbit anti-mouse Ig or HRP-conjugated rabbit anti-goat IgG, followed by analysis using the Lumino-Image Analyzer, Las 4000 Mini (GE Healthcare). Proteins on the membrane were stained with GelCode Blue (Thermo Fisher Scientific, Waltham, MA, USA), SilverQuest Silver Stain (Invitrogen), or SYPRO^®^ Ruby Protein Gel Stain (Invitrogen) according to the manufacturers’ protocols. 

### 2.5. Identification of the R-10G-Binding Protein in 201B7 Cells 

Following SDS-PAGE of the purified R-10G-binding protein, the immunoreactive protein bands were excised from the gel and subjected to in-gel digestion. The peptides released were analyzed by LC-MS/MS using a hybrid quadrupole-Orbitrap mass spectrometer (Q-Exactive, Thermo Fisher Scientific) interfaced online with a nano-flow HPLC (EASY-nLC 1000, Thermo Fisher Scientific). The sample was loaded onto the trap column (0.075 mm i.d. × 20 mm, 3 μm, Acclaim PepMap 100, Thermo Fisher Scientific), and separation was performed on a C18 column (0.075 mm i.d. × 125 mm, 3 μm, NTCC-360/75-3-125, Nikkyo Technos, Tokyo, Japan) at a flow rate of 300 nL/min. The eluents consisted of 0.1% formic acid (pump A) and 80% CH_3_CN and 0.1% formic acid (pump B), and peptides were eluted using a linear gradient of 0–35% B. Data-dependent MS/MS acquisitions were performed for the most intense ions as precursors. Proteins were identified by searching the human protein database (UniProt) using the Discoverer v1.4 search engine (Thermo Fisher Scientific).

### 2.6. Chemical Analysis of R-10G-Binding Protein in 201B7 Cells

Determination of sialic acid content in R-10G-binding protein from 201B7 cells was performed as described previously [13]. R-10G-binding protein (5 µL, 8.5 ng) in 50 µL 50 mM hydrochloric acid was heated at 80 °C for 1 h. The hydrolysate was applied to centrifugal ultrafiltration membranes. The flow through (20 µL) was evaporated, and the sample was resuspended in 7.5 µL 75% acetonitrile. A 5 µL aliquot of the sample solution was subjected to HPLC using an InertSustain Amide Column (GL Sciences, Tokyo, Japan).

Amino sugars were analyzed as described previously [14]. Briefly, samples were subjected to hydrolysis in 6 N HCl at 100 °C for 2.5 h. Amino sugars released by hydrolysis were separated on a TSK gel SCX column (4.6 mm i.d. × 150 mm) and eluted with 0.35 M borate/NaOH buffer (pH 7.6) at 60 °C, with a high sensitivity achieved by a post-column reaction with 1% 2-cyanoacetamide. The oligosaccharides released from keratan sulfates upon keratanase II digestion or endo-β-galactosidase digestion were separated using a gel permeation HPLC system with fluorometric post-column detection [6].

## 3. Results

### 3.1. Glycan-Epitope Profiles of 201B7 Cells

We examined the glyco-epitope profiles of the crude extracts of 201B7 cells, as described previously in Tic cells, which were used as antigens to raise anti-iPSCs in mice [6,8]. Upon Western blotting, R-10G, R-17F, and TRA-1-60 yielded a single major band at ~250 kDa, similar to the result using an anti-podocalyxin antibody (Figure 1A). In addition to a single major band at 250 kDa, R-17F yielded second and third bands at ~100–150 and 75 kDa. Gel Code Blue (protein) staining (Figure 1B) revealed a large number of bands between 20 and 150 kDa, in addition to two bands at ~250 kDa, in the crude extracts of 201B7 cells. Therefore, the glycoproteins carrying these glycan epitopes (R-10G, R-17F, and TRA-1-60) might be present as high-molecular-weight proteins on the surface of hiPSCs (201B7).

The glyco-epitope profiles of 201B7 cells were similar to those in our previous reports on Tic hiPSCs [6,8]. In addition, they were in agreement with the co-expression profiles of these epitopes revealed by flow cytometry and by confocal laser microscopy using the same antibodies ([6,8], Appendix A). Therefore, these epitopes are shared by hiPSCs. However, it is of note that the expression of R-10G epitope was higher in Tic cells than in 201B7 cells (Figure 1A, lane 1 vs. 2, in which equal amounts of total cell proteins (6 µg) were loaded). 

### 3.2. Purification of the R-10G-Binding Protein from 201B7 Cells

Next, we purified the R-10G-binding protein from 201B7 cells, as described for purification of R-10G from Tic cells. The 201B7 cell extracts were loaded onto an R-10G Sepharose 4B affinity column, and the bound proteins were eluted with 0.1 M diethylamine-HCl (pH 11.5) [6]. After resolution by SDS-PAGE, proteins were subjected to Western blotting using R-10G. All R-10G-binding proteins applied bound to the column and were eluted as a single major band at ~250 kDa (tubes 15–20, Appendix A). However, this one-time purified R-10G-binding protein yielded numerous protein bands at 37–150 kDa (data not shown) by silver staining, indicating that additional purification was required. Re-chromatography of the one-time purified R-10G-binding protein on the same R-10G Sepharose 4B column resulted in its complete retention and almost quantitative recovery of the eluate fraction (tubes 16–21, Appendix A). After SDS-PAGE and silver staining, this two-time purified R-10G-binding protein yielded a single predominant protein band (Figure 2A, lane 1) and a predominant immunoreactive band with R-10G (lane 2) also at ~250 kDa, indicating the R-10G-binding protein to be of high purity.

SDS-PAGE of the purified R-10G-binding protein, followed by Western blotting using the R-10G, R-17F, and TRA-1-60 antibodies, yielded a single major band at ~250 kDa, similar to that yielded using the anti-podocalyxin antibody (Figure 2B), suggesting that the R-10G, R-17F, and TRA-1-60 glycan epitopes are present on a common core protein, podocalyxin. 

Next, purified R-10G-binding protein was subjected to blue native PAGE and Western blotting using R-10G and R-17F antibodies. Blue native PAGE is superior to SDS-PAGE for separating large membrane proteins [15]. The purified R-10G-binding protein from 201B7 cells yielded a single broad band at 480–1236 kDa by protein staining (Figure 2C, lane 1) and Western blotting using R-10G (Figure 2C, lane 2) or R-17F (Figure 2C, lane 3). Note that the major parts of these two epitopes overlapped along the broad bands, but R-10G staining was stronger in the upper half of the band and R-17F staining in the lower half. Therefore, the purified R-10G-binding protein comprised a family of proteins bearing the R-10G and R-17F epitopes. The larger proteins contained more R-10G epitopes and the smaller proteins more R-17F epitopes. 

### 3.3. Identification of the Purified R-10G-Binding Protein as Podocalyxin

The above results suggested that the R-10G-binding protein purified from 201B7 cells is a near-homogeneous family of highly purified proteins. Purified R-10G-binding protein was subjected to SDS-PAGE and Western blotting using R-10G and R-17F antibodies and staining with SYPRO Ruby (Figure 3A).

R-10G and R-17F immunostaining and SYPRO Ruby staining indicated a broad band at ~250 kDa, although each lane consisted of several bands that did not completely overlap. Therefore, we divided these areas into three sections. Band “a” was at ~250 kDa and showed the same positivity for R-10G, R-17F, and SYPRO Ruby staining. Band “b” was above 250 kDa and was strongly positive for R-10G but less so for R-17F. Band “c” was just below 250 kDa, which was positive only for R-17F. The gels were excised into three fractions (a, b, c; SYPRO Ruby staining; Figure 3A) and subjected to in-gel digestion followed by LC-MS/MS (Figure 3B). The three fractions of the major Western blot band generated several peptide sequences, all of which corresponded to partial sequences of podocalyxin. Also, podocalyxin was the only protein common to bands a–c and had the highest coverage of proteins from bands a and b. 

Podocalyxin is the major sialoprotein expressed on podocytes from rat kidney glomerulus and has a molecular weight of 140 kDa [16]. Human podocalyxin is a type 1 transmembrane protein belonging to the CD34 family of sialomucins (160–165 kDa) and its gene encodes a protein of 528 amino acids. The extracellular domain of podocalyxin is extensively glycosylated with sialylated O-linked carbohydrates and five potential sites for N-linked glycosylation [17,18,19]. Podocalyxin is expressed primarily in vascular endothelia of adult vertebrates [20] and is required for maintaining the integrity of the blood–brain barrier [21]. Podocalyxin is also a remarkable glyco-epitope carrier. The human stem cell marker epitopes TRA-1-60 and TRA-1-81 are contained within podocalyxin [22]. hiPSC-derived podocalyxin carries not only R-10G and R-17F [6,9] but also additional epitopes recognized by newly generated anti-hiPSC antibodies (R-6C and R-13E; manuscripts in preparation). 

### 3.4. Characterization of the Glycans of Purified R-10G-Binding Protein by Glycosidase Digestion

Next, we digested the purified R-10G-binding protein with glycosidases prior to SDS-PAGE and determined the effects of digestion on the intensities and migration positions of the immunoreactive bands.

First, we digested purified R-10G-binding protein with PNGase F prior to SDS-PAGE, which released N-linked glycans from the core protein. The digestion resulted in essentially no decrease in R-10G-binding activity (Figure 4A, lane 1), indicating that N-linked glycans are not the major epitopes. However, PNGase F digestion caused a shift of the predominant band from the middle to lowest position of the broad band, indicating the presence of a considerable number of N-linked glycans lacking R-10G-binding activity. Therefore, the glycan epitopes are likely present only on O-linked glycans.

Next, we characterized the R-10G epitope by applying glycosaminoglycan-degrading enzymes. Chondroitinase ABC (lane 2), which degrades the chondroitin sulfate subfamily, and a heparinase mixture (lane 3), which degrades various subtypes of heparan sulfates and heparins, did not decrease the R-10G-binding activity (Figure 4A). Indeed, reactivity was enhanced. Therefore, neither heparan sulfate/heparin nor chondroitin sulfates are major constituents of the epitope structure. Digestion of the R-10G-binding protein with either α1-3/4 fucosidase or α1-2 fucosidase did not yield a detectable change in the immunoreactive bands, excluding a major role of fucose residues in the R-10G epitope (lanes 4 and 5). Similarly, digestion with neuraminidase (lane 6) did not decrease binding activity, excluding a role of neuraminic acid.

However, keratanase (lane 7), which digests keratan sulfate when the C-6 of GlcNAc is sulfated but the C-6 of galactose is not [23], significantly decreased the R-10G-binding activity, suggesting that the epitope harbors a keratan sulfate. We next applied two other keratan sulfate-degrading enzymes: keratanase II (hydrolyzes the 1,3-β-glucosamine linkages in keratan sulfate to galactose when the 6-O-position of GlcNAc is sulfated [7,24]) and endo-β-galactosidase (hydrolyzes the 1,4-β-galactosidic linkage when the 6-O-position of the Gal residue is not sulfated, irrespective of the presence of sulfate on the adjacent GlcNAc) [23,25]. As shown in Figure 4B, the Western blot band representing the R-10G-binding protein from 201B7 cells disappeared completely after digestion with endo-β-galactosidase (lane 1) but was merely reduced in intensity after digestion with keratanase II (lane 3). This suggests that the R-10G epitope on the R-10G-binding protein from 201B7 cells contains a few sulfated sugars. Similar results were obtained when the R-10G-binding protein from Tic cells was treated with these two glycosidases (lanes 2 and 4, respectively). However, the R-10G band at >250 kDa protein disappeared after keratanase II digestion (lane 4). It is possible that a part of the R-10G-binding protein in this high-molecular-weight region was more sensitive to keratanase II digestion, likely because of a relatively high local frequency of sulfated sugars.

### 3.5. Chemical Analysis of the R-10G Epitope 

We next determined the sialic acid content of the R-10G-binding protein purified from 201B7 cells by HPLC (Figure 5A). This method involved separation and detection of N-acetylneuraminic acid (Neu5Ac) and N-glycolylneuraminic acid (Neu5Gc) by hydrophilic interaction liquid chromatography and a fluorometric post-column reaction using 2-cyanoacetamide [13]. The purified R-10G-binding protein, following hydrolysis in 50 mM HCl at 80 °C for 1 h, showed a single peak at the position of Neu5Ac and none at the position of Neu5Gc, indicating Neu5Ac to be the predominant constituent of the protein, in which it is present at 3.98 × 10^−12^ mol/ng protein.

We also determined the amino sugar contents of the R-10G-binding protein purified from 201B7 cells (Figure 5B). Lyophilized proteins were hydrolyzed in 6 M HCl and heated at 100 °C for 2.5 h. The sample was subjected to hydrophobic interaction liquid chromatography using a TSK gel SCX column, and eluates were detected by fluorometric post-column reaction using 2-cyanoacetamide. GlcN (peak 1) and GalN (peak 2) were effectively separated, and the GlcN and GalN contents of R-10G-binding protein were 15.7 and 3.92 × 10^-12^ mol/ng protein respectively, with a GlcN/GalN molar ratio of 4.00. Based on the sialic acid content of the protein, the GlcN/Neu5Ac molar ratio was 3.95.

Based on the different sensitivities to keratanase II digestion of the R-10G-binding proteins isolated from 201B7 and Tic cells (Figure 4B, lanes 2 and 4), we conducted a chemical analysis of R-10G-binding protein purified from 201B7 cells. In our prior keratan sulfate analysis, the oligosaccharides released by keratanase II digestion were determined by reversed-phase ion-pair chromatography using a fluorometric post-column detector [6]. Here, we used a new method involving gel permeation chromatography using a fluorometric post-column detector [7] and digestion with endo-β-galactosidase rather than keratanase II. This enabled analysis of keratan sulfate oligosaccharides with low or no sulfated sugars (i.e., polylactosamine).

We first evaluated the procedure using bovine cornea keratan sulfate. As shown in Figure 5C(a), after endo-β-galactosidase digestion, GlcNAc(6S) β1-3Gal (peak 1) was obtained as the predominant oligosaccharide and GlcNAcβ1-3Gal (peak 2) was also detected as a minor component, in agreement with a prior report [26].

Then, endo-β-galactosidase digests of the R-10G-binding protein isolated from 201B7 cells were analyzed by HPLC (Figure 5C(b)). Peak 1 corresponding to GlcNAc(6S) β1-3Gal was predominant, as expected, and peak 2 corresponding to GlcNAcβ1-3Gal was also detected significantly. The GlcNAc(6S) β1-3Gal and GlcNAcβ1-3Gal contents of R-10G-binding protein were estimated to be 1.18 and 0.443 × 10^−12^ mol/ng protein respectively, with a molar ratio of 2.67. Therefore, approximately 27% of keratan sulfate-type oligosaccharides had a polylactosamine-type structure. We also analyzed keratanase II digests of the R-10G-binding protein from 201B7 cells (Figure 5D(b)). A peak corresponding to Gal β1-4GlcNAc(6S) (peak 2) again predominated, with a negligible peak corresponding to Gal(6S) β1-4GlcNAc(6S) (peak 1). The Gal β1-4GlcNAc(6S) content of R-10G-binding protein was estimated to be 0.388 × 10^−12^ mol/ng protein. Keratanase II digests of corneal keratan sulfate showed comparable levels of Gal β1-4GlcNAc(6S) (peak 2) and Gal(6S) β1-4GlcNAc(6S) (peak 1) under the same conditions (Figure 5D(a)). The above results are summarized in Table 1. R-10G-binding protein consists of a family of different keratan sulfates with relatively low sulfate contents, the majority of which consisted of GlcNAc(6S) β1-3Gal oligosaccharides with no additional sulfate residues. Notably, the amount of GlcN (4.00) far exceeded that expected merely from a component of keratan sulfate (0.11 + 0.30), suggesting a considerable portion of GlcN to be derived from core 3-type O-glycans (GlcNAc β1-3GalNAc). The amount of keratanase II digest (Galβ1-4GlcNAc(6S); 0.10) was comparable but not identical with that of endo-β-galactosidase digest (GlcNAc(6S) β1-3Gal; 0.30). The underlying reasons are unclear but may be linked to the structural complexity of these glycans and/or differences in experimental conditions. 

## 4. Discussion

The R-10G-binding protein purified from 201B7 cells was identified as podocalyxin by LC-MS/MS, just like the case of Tic cells, as was expected, confirming that podocalyxin is a predominant glyco-epitope carrier on the cell surface of hiPSCs. Glycans attach to proteins (glycoproteins) and also to lipids (glycolipids). Glycoproteins are comprised of N-linked glycans, O-linked glycans, and glycosaminoglycans (GAGs). Interestingly, the predominant portion of these glyco-epitopes on podocalyxins isolated from Tic cells and 201B7 cells appear to be expressed on O-linked glycans and/or glycosaminoglycans and not on N-linked glycans, although 5 potential N-glycosylation sites are present in the mature protein. Most of the R-17F epitopes appear to be expressed on O-linked glycans. On the other hands, the R-10G epitope appears to be on keratan sulfate. Keratan sulfate is a member of GAGs. GAGs are polysaccharides composed of negatively charged repeating disaccharide units. They are classified on the basis of structure into several groups such as chondroitin sulfate, heparin, heparan sulfate, hyaluronan (not sulfated), and keratan sulfate. Keratan sulfate disaccharide unit is Galβ1-4GlcNAc β1-3 and this backbone is almost always 6-O-sulfated on GlcNAc and, to a variable extent, on Gal. Keratan sulfate was initially identified in bovine cornea, followed by in cartilage, and a current active area of keratan sulfate research involves phosphocan in the central nervous system [27]. The degree of sulfation at the C-6 position of the galactose residues differed among these samples in the following order: bovine cartilage > bovine cornea > brain > hiPSC (Tic) > hiPSC (201B7) [28]. Chemical analyses have revealed that the R-10G-binding protein of 201B7 cells contained a polylactosamine disaccharide unit (Galβ1-4GlcNAc) at a significant frequency. This is in contrast with the R-10G-binding protein derived from Tic cells, in which keratan sulfate disaccharides units mostly consisted of Galβ1-4GlcNAc(6S). This finding is an example of structural micro-diversity among hiPSCs-glycans. These structural differences may affect the binding activity of R-10G to hiPSCs, because we showed previously that R-10G recognizes Galβ1-4GlcNAc(6S)β1-3Galβ1-4GlcNAc(6S)β1 as a minimum epitope, but it does not recognize Galβ1-4GlcNAcβ1-3 Galβ1-4GlcNAc β1 at all [7]. Indeed, in a semi-quantitative analysis (Appendix A), the expression level of the R-10G epitope derived from 3 × 10^4^ 201B7 cells (lane 1) was most comparable with that derived from 3 × 10^3^ Tic cells (middle in the lane 2), indicating that the amount of R-10G epitope present in one 201B7 cell was significantly lower than that in the Tic cell (Appendix A). On the other hand, the level of podocalyxin derived from 3 × 10^4^ 201B7 cells (Appendix A, lane 1) was most comparable with that derived from 3 × 10^4^ Tic cells (middle in the lane 2). Thus, the number of podocalyxin molecules expressed on one cell appears to be similar between 201B7 and Tic cells. It would be interesting to know whether the Galβ1-4GlcNAc structure is distributed at random along the keratan sulfate chain or localized at some specific points in the chain.

The R-17F epitope, another major epitope on the R-10G-binding protein, is proposed to be a blood group H type 1 oligosaccharide, Fucα1-2Galβ1-3GlcNAcβ1-3Gal. When 201B7 cell lysates were separated on an R-10G Sepharose 4B affinity column, a considerable amount of R-17F epitope, but not R-10G epitope, was recovered in the flow-through fraction. Western blotting of this fraction using R-17F showed one major band at ~250 kDa, similar to that of the anti-podocalyxin antibody (data not shown). Therefore, a considerable level of podocalyxin harbors an R-17F epitope but no R-10G epitope. These results are consistent with the semi-quantitative analysis of R-10G epitope expression described above and our prior co-expression studies of these epitopes on Tic cells [6] and also on 201B7 cells (Appendix A). The R-17F epitope is expressed strongly all over the cell membranes by a variety of types of hiPSCs and thus has potential as a marker of hiPSCs/hESCs, but expression of the R-10G epitope varies among cells even within the same colony.

In the last decade, glycomic profiles of hPSCs/hESCs have been analyzed extensively, mainly by mass spectrometry [28,29]. However, very little light has been shed on keratan sulfate present on hiPSCs/hESCs. Several years ago, on the basis of the binding specificity of R-10G, we proposed that keratan sulfate lacking over-sulfated structures may be specifically expressed on the surface of hiPSCs/hESCs [30,31]. Very recently, Wu et al. assigned Galβ1-4GlcNAc(6S)β1-3Galβ1-4GlcNAcβ1-3Galβ1-4GlcNAc as an R-10G epitope structure in bovine cornea keratan sulfate by a Beam Search approach [32]. At this moment, we have an ongoing project to analyze the whole glycans of the R-10G binding protein isolated from 201B7 cells in collaboration with a top scientist in the field of sulfated glycan analysis. This approach is expected to provide us with very useful information on the subject of glycan diversity, including the real sequences of epitope glycans, epitope locations on the glycan chains, frequencies of their appearances, and presence or absence of branching structures, as well as the number of sulfates on the glycan epitopes.

## 5. Conclusions

Podocalyxin was identified as the predominant glycan-epitope carrier protein common to hiPSCs.

Keratan sulfate-type epitopes were shared by two hiPSC types. The keratan sulfate from 201B7 cells contained a polylactosamine disaccharide unit (Galβ1-4GlcNAc) at a significant frequency, but that from Tic cells consisted mostly of keratan sulfate disaccharides (Galβ1-4GlcNAc(6S)). In addition, the abundance of R-10G epitope glycans was significantly lower in 201B7 cells than that in Tic cells.

## Figures and Tables

**Figure 1 biomolecules-11-00508-f001:**
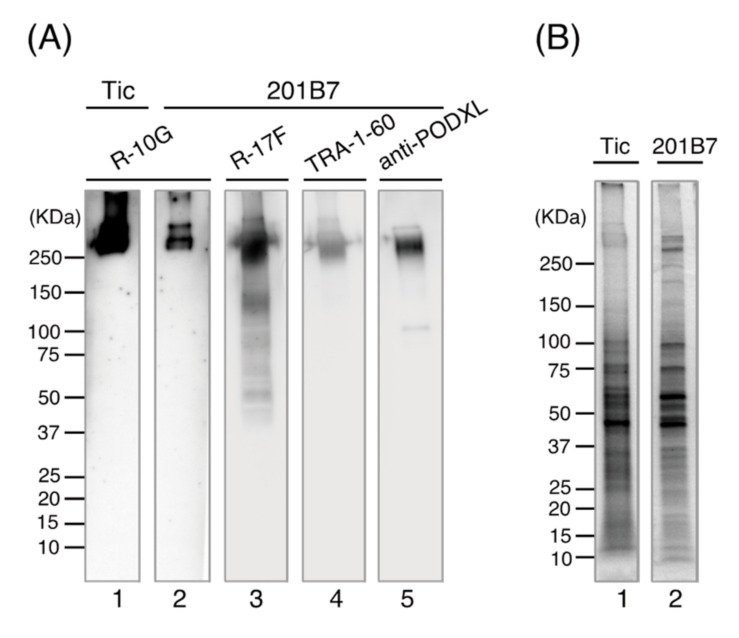
SDS-PAGE of iPSC lysates and Western blotting using the indicated antibodies. (**A**) Whole-cell lysates of 201B7 cells (8.0 × 10^4^) and Tic cells (6.2 × 10^4^) in complete RIPA buffer were resolved by SDS-PAGE on a 4–15% gradient gel under reducing conditions. The proteins were Western blotted using R-10G, R-17F, TRA-1-60, and anti-podocalyxin antibodies (3 µg protein/mL). (**B**) Gel Code Blue-stained gel of 201B7 (7.7 × 10^4^) and Tic (5.2 × 10^4^) cell lysates.

**Figure 2 biomolecules-11-00508-f002:**
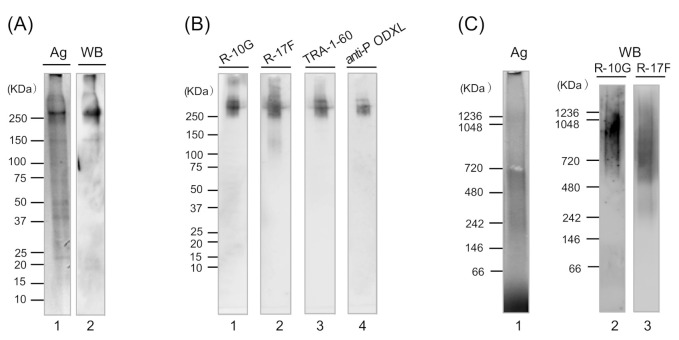
SDS-PAGE and blue native PAGE of R-10G-binding protein purified from 201B7 cells. (**A**) Purified R-10G-binding protein from 201B7 cells (1 ng protein) was resolved by SDS-PAGE on a 4–15% gradient gel under non-reducing conditions, followed by silver staining (lane 1) or Western blotting using R-10G (3 µg protein/mL) (lane 2). (**B**) The purified R-10G-binding protein from 201B7 cells (0.25 ng protein) was resolved by SDS-PAGE on a 4–15% gradient gel under non-reducing conditions, followed by Western blotting with the following antibodies (3 µg protein/mL): R-10G (lane 1), R-17F (lane 2), TRA-1-60 (lane 3), and anti-podocalyxin (lane 4). (**C**) R-10G-binding protein purified from 201B7 cells (1 ng protein in lane 1, 0.3 ng in lanes 2 and 3) was resolved by blue native PAGE on a 3–12% gradient gel under non-reducing conditions, followed by silver staining (lane 1) or Western blotting using antibodies (3 µg protein/mL) against R-10G (lane 2) and R-17F (lane 3).

**Figure 3 biomolecules-11-00508-f003:**
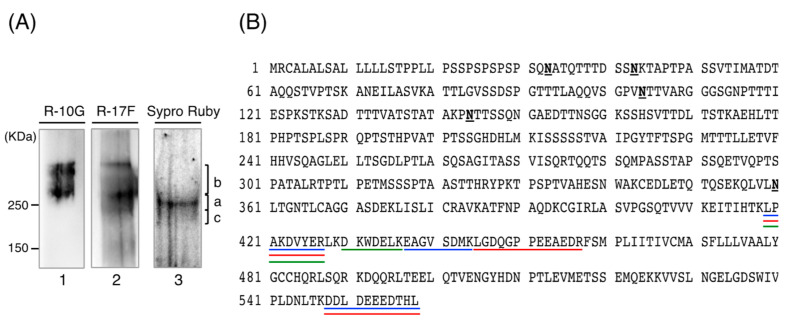
Identification of the R-10G-binding protein as podocalyxin by LC-MS/MS. (**A**) Western blotting using R-10G and SYPRO Ruby staining. Purified R-10G-binding protein (3 ng protein in lanes 1 and 2, 10 ng protein in lane 3) was resolved by SDS-PAGE on a 4–15% gradient gel under reducing conditions. Lanes 1 and 2 show the results of Western blotting using R-10G and R-17F respectively, and lane 3 shows SYPRO Ruby staining. The protein bands a, b, and c were excised from the gel and analyzed by LC-MS/MS. Protein concentrations were estimated based on the SYPRO Ruby staining intensity in comparison with two concentrations of standards from the High-Molecular-Weight SDS Calibration Kit for Electrophoresis (Amersham Pharmacia Biotech). (**B**) Identification of R-10G antigen by LC-MS/MS. The identified peptides in band “a” are underlined in blue within the complete human podocalyxin sequence. Those in band “b” are underlined in orange and those in band “c” in green. N, potential N-glycosylation sites.

**Figure 4 biomolecules-11-00508-f004:**
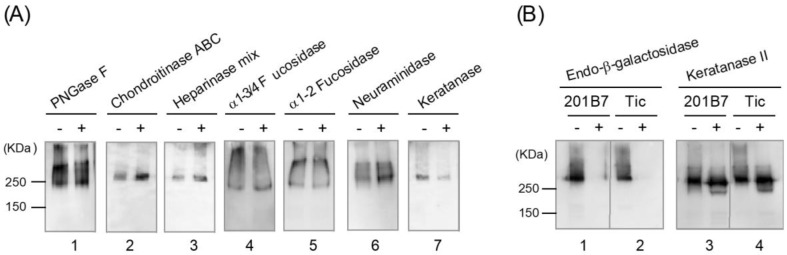
Characterization of the purified R-10G-binding protein by glycosidase digestion and Western blotting. Purified R-10G-binding protein (0.5–4 ng) incubated with (+) or without (−) glycosidases was subjected to SDS-PAGE on a 4–15% gradient polyacrylamide gel under non-reducing conditions. The exception was PNGase F digestion, which requires preheating under reducing conditions. Western blotting using R-10G was subsequently conducted. (**A**) Purified R-10G-binding protein from 201B7 cells was digested with PNGase F (lane 1), chondroitinase ABC (lane 2), a heparinase mixture (lane 3), α1-3/4 fucosidase (lane 4), α1-2 fucosidase (lane 5), neuraminidase (lane 6), and keratanase (lane 7), and the digests were analyzed by Western blotting using R-10G. (**B**) Purified R-10G-binding protein from 201B7 (lanes 1, 3) and Tic (lanes 2, 4) cells incubated with (+) or without (−) endo-β-galactosidase (lanes 1, 2) or keratanase II (lanes 3, 4) was analyzed by Western blotting using R-10G.

**Figure 5 biomolecules-11-00508-f005:**
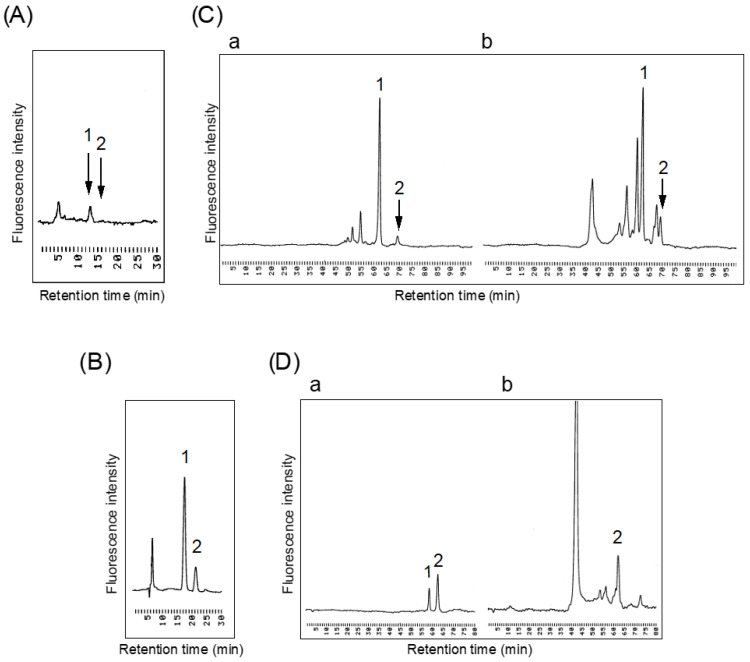
Chemical analysis of R-10G-binding protein. (**A**) HPLC analysis of sialic acid in purified R-10G-binding protein. Peak 1, NeuAc; peak 2, NeuGc. (**B**) HPLC analysis of amino sugars in R-10G-binding protein. Peak 1, GlcN; peak 2, GalN. (**C**) HPLC analysis of oligosaccharides released by endo-β-galactosidase digestion. Peak 1, GlcNAc(6S) β1-3Gal; peak 2, GlcNAcβ1-3Gal. (a) Bovine cornea keratan sulfate, (b) R-10G-binding protein. (**D**) HPLC analysis of oligosaccharides released by keratanase II digestion. Peak 1, Gal(6S) β1-4GlcNAc(6S); peak 2, Galβ1-4GlcNAc(6S). (a) Bovine cornea keratan sulfate, (b) R-10G-binding protein.

**Table 1 biomolecules-11-00508-t001:** Chemical analysis of R-10G-binding protein from 201B7 cells.

Carbohydrate	Molar Ratio
GalN ^(a)^	1.00
GlcN	4.00
Neu5Ac	1.01
GlcNAcβ1-3Gal	0.11
GlcNAc(6S) β1-3Gal	0.30
Galβ1-4GlcNAc(6S)	0.10

^(a)^ The amount of GalN is defined as 1.00.

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
