# Peer review of "Glycan Epitopes on 201B7 Human-Induced Pluripotent Stem Cells Using R-10G and R-17F Marker Antibodies"

_biomolecules, 2021, doi:10.3390/biom11040508_

Round 1
Reviewer 1 Report
Authors improved their paper.
Reviewer 2 Report
The authors have answered the questions in a satisfactory manner,
This manuscript is a resubmission of an earlier submission. The following is a list of the peer review reports and author responses from that submission.
Round 1
Reviewer 1 Report
This manuscript is written about the production of new two single-clone antibodies bonding with hiPSC specific glycan. It revealed the fact that a variety of proteins (keratan sulfate-type epitope and lacto-N-fucopentaose-1) bonding with two antibodies are recognized.
The Carbohydraste-recognizing antibodies proposed in this manuscript are helpful to recognize specific glycan structures in hiPS. However, the manuscript fails to clarify differences from conventional pluripotent stem cell specific antibodies, such as SSEA3/4 or TRA1-60, and the advantages of the proposed ones.
If the difference is that conventional antibodies bond in hECC and R-10G/R-17F bonds in hiPSC and hESC only, it is necessary to suggest experimentally proved results in detail.
This study argues that two single-clone antibodies can recognize iPSC specifically by using two iPSC lines. The argument, however, is not enough.
The authors need to prove their argument by using multiple types of iPSC ( the iPSC generated in a different way) and hESC cells. The iPSC generated by retrovirus is likely to cause deformation of glycan epitopes so that it is unable to be used universally.
In order to prove the hypotheses made in this study, authors need to make verification with a variety of pluripotent stem cells.
Authors need to provide more general information in introduction session, and to explain in more detail the value of newly revealed information in discussion session.
Reviewer 2 Report
Authors should add some morphological experiments to ameliorate the paper presentation and design.
Reviewer 3 Report
The manuscript entitled "Glycan Epitopes on 201B7 Human Induced Pluripotent Stem Cells using R-10G and R-17F Marker Antibodies" investigates and characterizes the epitopes R10G and R17F on the surface of the specific iPSC line. The chemical analysis and experimental design were very scientifically sounded. However, there are multiple weakness in the manuscript including the major technicity of the writing.
- What is the application and outcome of the study results. It would have been much appreciated if the results of this study are applied to show the benefits of the study results. What is the specific goal or novelty that this study investigates? Please add more discussion about the impact of this research and also some background information in the introduction.
- Majority of the study involves immunological analysis and analytical chemistry, is this manuscript satisfy the goals of this journal?
- Why has the authors used only one cell like of iPSCs, multiple iPSC lines should be investigated as well. what is the passage number when the iPSCs were used? Does the source of the iPSCs affect the result of the study?
Conclusion: The study as well as its application should be reevaluated prior to publication.